# Bio-Based Coatings for Food Metal Packaging Inspired in Biopolyester Plant Cutin

**DOI:** 10.3390/polym12040942

**Published:** 2020-04-18

**Authors:** José J. Benítez, Sonja Osbild, Susana Guzman-Puyol, Antonio Heredia, José A. Heredia-Guerrero

**Affiliations:** 1Instituto de Ciencia de Materiales de Sevilla, Centro Mixto CSIC-Universidad de Sevilla, Americo Vespucio 49, Isla de la Cartuja, E-41092 Sevilla, Spain; sonja.osbild@hotmail.de; 2Instituto de Hortofruticultura Subtropical y Mediterránea (ISM)-La Mayora, Departamento de Mejora Genética y Biotecnología (CSIC), Algarrobo-Costa, E-29750 Málaga, Spain; susana.guzman@csic.es (S.G.-P.); heredia@uma.es (A.H.); ja.heredia@csic.es (J.A.H.-G.)

**Keywords:** bio-based polymers, fatty polyhydroxyesters, can coatings, esterification kinetics, melt-polycondensation

## Abstract

Metals used for food canning such as aluminum (Al), chromium-coated tin-free steel (TFS) and electrochemically tin-plated steel (ETP) were coated with a 2–3-µm-thick layer of polyaleuritate, the polyester resulting from the self-esterification of naturally-occurring 9,10,16-trihydroxyhexadecanoic (aleuritic) acid. The kinetic of the esterification was studied by FTIR spectroscopy; additionally, the catalytic activity of the surface layer of chromium oxide on TFS and, in particular, of tin oxide on ETP, was established. The texture, gloss and wettability of coatings were characterized by AFM, UV-Vis total reflectance and static water contact angle (WCA) measurements. The resistance of the coatings to solvents was also determined and related to the fraction of unreacted polyhydroxyacid. The occurrence of an oxidative diol cleavage reaction upon preparation in air induced a structural modification of the polyaleuritate layer and conferred upon it thermal stability and resistance to solvents. The promoting effect of the tin oxide layer in such an oxidative cleavage process fosters the potential of this methodology for the design of effective long-chain polyhydroxyester coatings on ETP.

## 1. Introduction

Canned food production is a global industry with a predicted turnover of 118 billion USD by 2023 and an estimated growth of 3.8% in the 2018–2023 period [1]. Because of their airtightness and high mechanical resistance, cans are a suitable format for long-shelf life and the long-distance logistics of ready-to-eat meals, vegetables, meat and seafood. The sector faces the challenge of providing chemical-free nutrients, and intensive research is being carried out to prevent the contamination of food with harmful substances stemming from the container. One substance that has triggered concern about canned food safety is bisphenol A (BPA). This molecule was employed in the formulation of inner lacquers used to prevent direct contact between the metal body and the foodstuff; concerns about its negative effects on human health arose about a decade ago [2,3,4,5]. BPA is an endocrine disruptor that is considered to be detrimental for reproduction, development and metabolic and immunologic functions in humans [6]. The main pathway of BPA to enter the human body is by food and beverage intake [7,8], and many studies have linked elevated levels of BPA in urine with the consumption of canned products [9,10]. The migration of BPA from the can coating has been demonstrated, and instances in which it reached as much as a few hundreds of micrograms per kilogram of canned foodstuff have been reported [11,12]. Consequently, alternatives to BPA epoxy resins have been recently investigated, though, to date, none of them has proven to be competitive enough in terms of cost, corrosion resistance, fabrication, organoleptic properties and appearance to take over the market [13]. Among them, oleoresins have regained interest [14,15]. However, some drawbacks like poor corrosion resistance and adherence to metal remain to be addressed. Other options are acrylics and polyester resins, though the former are brittle and impart flavor to food, while the latter are prone to chemical attack by acidic media. Also, to be used with highly acidic foods, polyolefin coatings are being developed [16]. To exploit their advantages and to palliate deficiencies, material combinations have been tested. For instance, to correct the low corrosion resistance of polyesters, they have been laminated on top of a metal-coating adhesive primer layer [13]. At this point, it is evident that there is no option with the market share that BPA-based resins had in the past, and solutions are particularized considering product- and fabrication-specific requirements. And, certainly, safety issues associated with the potential migration of substances from these new formulations will be a topic in the years to come.

Our approach to design a nontoxic coating layer for cans is based on observations of the unique barrier properties of the plant cuticle. The cuticle is the outermost membrane that protects the epidermis of fruits, leaves and nonlignified stems of higher plants [17]. The main component of the cuticle is cutin (up to 80% *w*/*w*), a nontoxic, biodegradable, hydrophobic, insoluble and infusible amorphous biopolyester mostly made of interesterified C_16_ polyhydroxy acids. The goal is to confirm whether a synthetic C_16_ polyhydroxyester resembling cutin may be developed as an effective coating for food cans. For this purpose, a C_16_ polyhydroxyacid such as aleuritic acid (9,10,16-trihydroxyhexadecanoic acid) was polymerized. Our group has already obtained and characterized free-standing polymeric films from aleuritic acid [18,19]. The thermal and solvent stability of the obtained polyaleuritate are promising candidates for a can coating material to withstand contact with liquids and sterilization. The synthetic route is self-esterification in air, which is a direct and easily scalable method using no catalyst or hazardous solvents. Thus, a 2–3 µm thick layer of aleuritic acid was deposited and thermally cured on three common metals used in the canning industry, i.e., Aluminum (Al), electrochemically tin-plated steel (ETP) and chromium-coated tin-free steel (TFS)). The kinetics, as well as the chemical composition, texture and other properties of the polyaleuritate coatings obtained were studied and evaluated.

## 2. Materials and Methods

### 2.1. Coating Preparation

Al, TFS and ETP metal plates were kindly provided by AkzoNobel Packaging Coatings S.A. (Spain), but specific information about their composition and manufacturing process was missing. For instance, we ignored the type of steel matrix (L or MR) and tin coating (E or D) in the ETP substrates and the chromium plating (one or two steps) in TFS. Prior to use, plates were cleaned using glassware soap, rinsed thoroughly with deionized water and ethanol and dried at room temperature. Aleuritic (9,10,16-trihydroxyhexadecanoic) acid (Alfa Aesar, purity ≥ 95%) was dissolved in ethanol (Honeywell, purity ≥ 99.8%) at a concentration of 10 mg/mL and sprayed with an airbrush (0.5 mm nozzle) onto preheated (~100 °C) Al, TFS and ETP substrates. The volume of the sprayed solution was adjusted to achieve about 0.3 mg/cm^2^ on square metal pieces 4.5 × 4.5 cm^2^. The coated specimens were placed inside an air-forced furnace and heated at temperatures ranging from 140 °C to 200 °C for variable periods of time (from 5 min up to 120 min). The coating thickness was calculated from the weight difference between coated and noncoated samples, and was estimated to be about 2.5 µm. Adhesion was qualitatively evaluated by thumb nail scratching and adhesive tape tests. 

### 2.2. Textural Characterization

The texture of the samples was determined with a Topometrix Explorer AFM (Santa Clara, CA, USA) equipped with a large scale scanner (130 × 130 µm^2^) and soft-contact silicon nitride lever (Budget Sensors, k = 0.06 N/m). Higher resolution images were acquired with a Nanotec AFM (Madrid, Spain) using a 10 × 10 µm^2^ scanner working with a contact pyrex-nitride probe (NanoWorld, k = 0.08 N/m). Both scanners were calibrated in the X, Y and Z directions using commercial gratings (NT-MDT). AFM images were processed and analyzed using the WSxM software (Nanotec, Spain). For representativeness, two to three preparations per sample were imaged at four distant points using both the large and small scanners. 

### 2.3. Chemical Analysis

A chemical analysis of the coatings was performed by FT-IR spectroscopy using a specular reflectance accessory (Smart SpeculATR, Thermo Scientific) coupled to a Nicolet iS50 spectrometer equipped with a DLaTGS detector. The accessory was continuously purged with dry N_2_ to reduce the contribution of ambient CO_2_ and water. Specular reflectance is a very suitable mode for large areas and deep sampling of polymer coatings thicker than 1 µm on metals. In this case, the analysis area was ~3.1 cm^2^, which ensured high signal levels and sample representativeness. Fifty scans were accumulated at 4 cm^−1^ resolution and clean metal supports were used as a background. Signal intensity is quantitative when expressed as the logarithm of the inverse of the relative reflectance (log(1/R)). Data acquisition, processing and band fitting was performed with the OMNIC 9 (Thermo Scientific) software package.

### 2.4. Wettability and Solubility Measurements

Surface hydrophobicity was evaluated by means of static water contact angle (WCA) measurements using an Attension TL100 Optical Tensiometer (KSV, Helsinki, Finland) in sessile drop mode. A 3 µL Milli-Q grade water drop was deposited on the surface of samples and the contour was recorded for 30 s at 12 fps. The contact angle was measured on both sides of the drop contour and averaged. Frames with left and right values differing more than 2° were rejected. For reproducibility, up to five points of each sample were tested.

Coating resistance to solvents was determined by immersing the specimen in ethanol for 48 h under orbital agitation and recording the sample weight loss after drying with 0.1 mg precision. The results from three samples were averaged.

### 2.5. UV-Visible Reflection Spectra

UV-Vis total reflection spectra were obtained with a Cary 300 (Agilent, Santa Clara, CA, USA) spectrometer and using a integrating sphere with a 8° wedge (Labsphere). Clean metal substrates were used as references for nominal 100% total reflectance.

## 3. Results and Discussion

### 3.1. Texture and Roughness of Supports and Coatings

The texture of the samples was studied by both SEM (Hitachi S4800, Duesseldorf, Germany) and AFM. An additional EDX (Bruker-X Flash-4010, Berlin, Germany) analysis of the bare supports revealed that the Al was quite pure, with traces of Si and Fe (~0.3% atom/each) and a passivation layer of aluminum oxide. TFS is a Fe matrix with a thin chromium oxide layer (Cr 0.8% atom) and ETP is made of an iron matrix covered by a crust of Sn oxide (Sn 32% atom). In the latter case, SEM micrographs revealed the presence of some scattered uncoated pits (data not shown).

Figure 1 shows the large range (130 × 130 µm^2^) surface topography obtained by AFM. On the supports, the ripples left by polishing were visible as vertical lines. The corrugation is more pronounced on ETP, while on Al and, particularly, TFS, large holes contributing to the increment of the h_max_ parameter were detected (Table 1). The AFM textural analysis was extended to the homogeneous regions between ridges (small images in Figure 2). As can be observed, the high magnification images characterized the porous ring-like structure of aluminum oxide in Al, similar to the one reported for anodized Al_2_O_3_ [20], the granular pattern exhibited by chromium oxides in TFS [21] and the interconnected structures of SnO_2_ in ETP [22]. The combination of long- and short-range textural data allowed us to calculate a specific surface area factor (S_F_) which indicated that the available surface of supports had grown in the order ETP < TFS < Al; see Table 1.

When coated with a 2–3 µm thick polyaleuritate layer, the surface topography changed, depending on the spreading of the polyester phase on the metal support. In Figure 1, three patterns can be observed. On Al, polyaleuritate extended evenly, covering the surface asperities and forming a quite flat deposit. In contrast, on ETP, the trend was to build globular deposits with a height comparable to the average thickness of the polyaleuritate layer. TFS was a combination of both, though wide bumps built up, and the filling of the asperities of the bare metal was also observed. Such behavior can also be visualized from the evolution of the long-range surface roughness; see Table 1 and plot A in Figure 2. Compared to the bare support, the formation of the flat deposit on Al caused a reduction in the RMS value, while the generation of globular structures on ETP significantly increased the surface roughness. In TFS, the asperity filling was compensated for by the development of the bumps and both the uncoated and coated specimen showed a similar long-range roughness (RMS) value. 

Though the full coverage of Al and TFS supports could be inferred from the long-range textural analysis, it was not possible to assess whether this was the case for ETP. Line profiles (Figure 1) showed that the underlying texture of ETP was mostly unaltered by the coating. However, the size of the globules did not seem to be enough to account for the mass of a 2–3 µm thick flat coating and, consequently, the development of a very thin layer at the background regions of ETP was presumed. To further investigate this issue, a higher resolution AFM analysis was carried out on the background regions of both the coated and uncoated samples; see Figure 2. As observed, the short-range texture of Al and TFS changed dramatically upon coating. In both cases, an interconnected, needle-like structure developed, which was compatible with the build-up of a semicrystalline polyaleuritate film [19] that caused a notable increase in roughness. In ETP, the modification of texture was more subtle, and the interconnected aggregates observed on the bare metal were replaced by globular structures with a mild increment of roughness from 3.4 nm to 6.7 nm. Such structures suggested an amorphous polyaleuritate coating on the ETP.

The textural evolution of the coating is a complex process that results from the dynamic balance between surface-polyaleuritate and polyaleuritate-polyaleuritate interactions. In this sense, the hydrophilic forces between the oxide layer and the polar groups of aleuritic acid molecule, as well as the growing hydrophobic component arising from the formation of the polyaleuritate, should be considered. Thus, the formation of large aggregates can be associated with the rapid development of a hydrophobic polyaleuritate phase.

### 3.2. Comparative Chemical Characterization of Polyaleuritate Coatings

The coatings, produced at different temperatures and polymerization times, were characterized by specular reflectance FTIR. In Figure 3A, samples prepared at 200 °C for 10 min on the three supports are compared. The unreacted film of aleuritic acid on Al was included as a reference for the chemical modifications observed upon heating in air. The ester formation was indicated by the development of bands at 1177 and 1247 cm^−1^ corresponding to the ν(C–O–C), and ν(C=O) at 1733 cm^−1^. The peaks at 725 and 1465 cm^−1^ were due to the deflection modes of backbone methylene groups, while those at 1055 and 1060 cm^−1^ correspond to the deformation of (C–O) bonds of hydroxyls [23]. Specular reflectance FTIR also revealed some differences between the three preparations. The presence of a series of progression bands between 800 and 1300 cm^−1^ was an indication of crystallinity of the polyaleuritate film on Al and TFS. This result was consistent with the textural data provided by AFM (Figure 2). Conversely, the absence of such progression bands confirmed that the deposit on ETP was amorphous. The formation of a crystalline polyaleuritate phase was feasible because, in the reaction conditions used, the esterification of the primary hydroxyl to yield a linear polymer was favored [18,19].

In ETP, a band corresponding to carboxylate species (–COO^−^ at 1548 cm^−1^) suggested a specific chemical interaction between the coating and the tin oxide layer. Such an interaction was not readily observable on Al and TFS. Furthermore, small absorptions on the high wavenumber side of the carbonyl peak (1770 and 1800 cm^−1^), as well as peaks in the 1660–1670 cm^−1^, revealed the occurrence of oxidation and dehydration side reactions, respectively [24]. The low wavenumber side broadening of the ν(C=O) was due to the perturbation exerted by hydrogen-bonding and to the presence of unreacted acid molecules. 

### 3.3. Kinetic Analysis of the Esterification Reaction in Polyaleuritate Films on Metals

The progress of the self-esterification of aleuritic acid on the metal substrates was carried out by monitoring and extracting the components from the ν(C=O) region; see Figure 3B. The characteristic ester bands were those at 1733 and 1715 cm^−1^, corresponding to isolated and hydrogen-bonded carbonyls groups, respectively [25]. The unreacted aleuritic acid fraction was responsible for the contribution at 1700 cm^−1^.

The reaction progress (p) for the three supports, as a function of time and reaction temperature, is shown in Figure 4. As observed, conversion values depended on the support used; therefore, the metal background played an important active role in the reaction. In general, values increased in the order TFS < ETP < Al. In the following sections, the kinetics of the reaction is studied by using two different methods.

#### 3.3.1. Time-Dependent Method

In this method, the reaction progression (p) was analyzed according to the reaction time (t) at a constant temperature (T). Empirically, the best fit was obtained with a second order law (Figure 5):(1)p/(1−p)=2·k·t
where the rate constant (k) is the slope of the p/(1−p) vs t plots at every temperature. Rate constant values could be fitted to an Arrhenius equation:(2)$$k=A\cdot \exp\left(-E_{\mathrm{act}}/\mathrm{RT}\right)$$
where (A) is a pre-exponential factor, (R) the universal gas constant, (T) the absolute temperature and (E_act_) the activation energy of the reaction. The A and E_act_ values obtained from the ln k vs 1/T plots (Figure 5) are compiled in Table 2.

#### 3.3.2. Temperature-Dependent Method

With this procedure, instead of monitoring the reaction progression at a constant temperature, the samples were analyzed after a fixed reaction time at a varying temperature. Assuming the second order kinetics described above, Equation (1), and considering Equation (2):(3)$$p/\left(1-p\right)=2\cdot A\cdot \exp\left(-E_{\mathrm{act}}/\mathrm{RT}\right)\cdot t$$

Thus, for a fixed time t = t_o_ at a given temperature (T), p = p_oT_
(4)$$\ln\left(p_{\mathrm{oT}}/\left(1-p_{\mathrm{oT}}\right)\right)=\ln\left(2{\mathrm{At}}_{o}\right)-E_{\mathrm{act}}/\mathrm{RT}$$
and, therefore, the E_act_ value can be calculated from the ln (p_oT_/(1 − p_oT_)) vs 1/T plots (Figure 6).

The pre-exponential factors and activation energy values obtained by this second method are also included in Table 2. The reliability of tabulated A and E_act_ values was supported by the good agreement between results obtained by both the time- and temperature-dependent methods. The temperature-dependent method is faster because it requires the preparation of a lower number of samples, but demands knowledge of the empirical reaction order.

As observed in Table 2, there were significant differences between E_act_ values among the metal substrates used. In terms of the activation energy, the reactivity increased in the order Al < TFS < ETP. To the best of our knowledge, there is only one reference in the literature dealing with the kinetics of the esterification of aleuritic acid in molten state [26]. By fitting the reported esterification conversion values (p) in such a study to Equations (1) and (2), an E_act_ = 98.8 kJ/mol could be calculated. This estimation could be considered as the reference for the noncatalyzed melt self-esterification of aleuritic acid. The value was comparable to those reported for the uncatalyzed reaction between fatty acids and alcohols (78.6–87.1 kJ/mol) [27], adipic acid and hexamethylene glycol (84.1 kJ/mol) [28] and sebacic acid and glycerol (71.1 kJ/mol) [29] in solution. The data in Table 2 show that E_act_ obtained for polyaleuritate on Al, TFS and ETP were below these references, which suggested a catalytic activity of the substrate in the esterification reaction. In fact, the addition of catalysts such as tetra *n*-buthyl titanate (TBT) caused an E_act_ reduction to 64.5–69.2 kJ/mol in the homogeneous esterification of fatty acids and alcohols [27] and to 47.4–59.5 kJ/mol for the esterification of poly(alkylene) succinates [30]. Also, the use of a strong acid such as *p*-toluenesulfonic acid yielded an activation energy of 47.9 kJ/mol for the melt polycondensation of 12-hydroxystearic acid [31]. The catalytic effect was associated with the tin oxide layer on ETP and, to a lesser extent, the chromium oxide on TFS. Indeed, Al and Sn metal complexes and oxides were used as catalysts for the ROP of lactones and the esterification of fatty acids with methanol [32,33] and the activity of chromium and chromium mixed oxides has also been reported in the formation of fatty acid methyl esters (FAMEs) [34]. In particular, the effectiveness of tin-based catalysts has been reported by reducing the activation energy for the polyesterification from 84.1 kJ/mol to 33.9 kJ/mol [28].

Despite the highest E_act_ value of polyaleuritate on Al, the conversion values (p) were the highest within the series; see Figure 4. Textural data revealed the particular nanostructure of the Al support and its high specific surface (S_F_). Indeed, the S_F_ parameter and the pre-exponential factor (A) showed the same trend. Thus, when interpreting conversion values, both E_act_ and the availability of catalytic reactive sites should be considered.

### 3.4. Side Reactions along the Formation of the Polyaleuritate Film on Metals by Heating in Air

In addition to the esterification, other reactions were detected in the formation of the polyaleuritate film on Al, TFS and ETP, particularly at high temperature (Figure 7). Among them were oxidation, dehydration and proton transfer. The extent of these processes depended on the reaction time and the type of metal support used. Thus, the formation of –COO^−^ groups stabilized due to the saturation of surface basic sites. The band was particularly intense on ETP because of the higher basicity of the tin oxide layer. Dehydration was mild and progressively increased with reaction time on Al and TFS. However, on ETP, the process seemed to be catalyzed by SnO_x_ and the reaction rate increased noticeably. However, after reaching a maximum, the concentration of C=C groups decreased, very likely because of the occurrence of another reaction consuming hydroxyls (i.e., oxidation in air). In any case, the most relevant side process was the generation of oxidized species (CO_ox_) such as peroxyesters and diacylperoxides (bands around 1800 and 1770 cm^−1^). Their generation has been shown to accompany oxidative diol cleavage and further esterification that has a strong influence of the structure and physical properties of free-standing polyaleuritate films obtained by melt-polycondensation in air [19]. The most relevant effect of such a process is the occurrence of branching and densification in the ester bond polymeric framework. Both effects led to structure amorphization that increased the insolubility and infusibility of polyaleuritate. Thermal stability and solvent resistance are positive traits for coatings of metal food containers undergoing washing and sterilization protocols and coming into extended contact with foodstuff fluids. The generation of CO_ox_ species was particularly remarkable on ETP; consequently, it could be concluded that SnO_x_ acted as a catalyst for both the esterification and the oxidative diol cleavage, as reported for other esterification catalysts such as Ti(OiPr)_4_ [19]. The promotion of the oxidative diol cleavage by SnO_x_ explained the obtained amorphous polyaleuritate films on ETP, as observed from textural and FTIR data (Figure 2 and Figure 3).

### 3.5. Surface Wettability and Solubility of Polyaleuritate Coatings

The performance of the polyaleuritate coating as a barrier between the metal and a water-based medium were evaluated by water contact angle measurements (WCA). The WCA were sensitive to many factors, mainly to surface chemical composition and roughness. The values observed for bare supports were 65° for Al and around 80° for TFS and ETP, which were comparable to those reported in the literature [35,36,37,38,39]. After coating, the initial WCA (WCA_o_) value increased slightly, i.e., about 6° for Al and ~12° for TFS. On ETP, the value was essentially the same (Figure 8).

On bare metals, no drop spreading was observed, which indicated the absence of contamination and/or a negligible chemical interaction with water. After coating with the polyaleuritate layer, the spreading (characterized by the parameter δ_WCA_) was intense for TFS, while it was much more moderate for Al and ETP. Both the low esterification degree (Figure 4) and the porous texture (Figure 2) may have contributed to an intense interaction between water and the coating on TFS. On the other side, beside the similar texture, the higher ester conversion led to a lower spreading on p-Al. p-ETP was characterized by low spreading and moderate WCA_o_ values which were associated with the low roughness and the high esterification degree of the amorphous polyaleuritate layer developed on this support. On p-ETP, the oxidative diol cleavage reaction also contributed by consuming surplus polar hydroxyls from the ester network, as deduced from the reduction of δ_WCA_ at longer heating times (Figure 8).

The resistance of the polyaleuritate coating towards lixiviation was evaluated for samples prepared at 200 °C. Ethanol is a good solvent for aleuritic acid and has been used to extract nonreacted molecules and low molecular weight oligomers from the coating. In Figure 9A, the solubility (as a weight percentage loss) is plotted versus the nonesterified aleuritic acid fraction (1 − p) for the three supports used. Three patterns can be distinguished: (i) coatings with moderate esterification degree (p ≈ 60–70%) on Al and TFS (blue circle) obeyed the theoretical recovery line (dashed), (ii) those with lower conversion in p-Al and p-TFS (green circle) were almost completely removed, and (iii) the ETP series (red circle) was characterized by solubility values well below those expected from the extraction of the nonesterified fraction (1 − p). Thus, it could be concluded that the esterification degree was not the only factor conditioning the solubility of the polyaleuritate layer. 

Solubility values better correlated with the presence of oxidized species (CO_ox_) (Figure 9B). It was suggested that such species were responsible for the amorphization and reduction of the surface specific area. Furthermore, CO_ox_ were also an indicator of the side oxidative diol cleavage that led to the densification of the ester network and the reduction of free hydroxyl groups. All of them were positive features to improve the resistance of the polyaleuritate coating to attack from polar solvents, and to explain the decreasing trend in Figure 9B.

### 3.6. Light Absorption and Reflection of Polyaleuritate-Coated Metals

The interaction of the inner protective coating with visible light is irrelevant from the point of view of the preservation of food in metal containers. However, to retain the glossy aspect of the original metal surface is an important esthetic aspect contributing to customer satisfaction. For this reason, the reflectivity of the coated samples was measured. Spectra of the series formed at 200 °C for 20 min are shown in Figure 10.

At the visible region (400–800 nm), the coating reduced the surface total reflectance by about 33% for p-TFS, 31% for p-ETP and only 8% for p-Al, which was visually acceptable. In the UV region (200–320 nm), the absorption of carbonyls [40] (bonding to antibonding, π→π*, around 200 nm and nonbonding to antibonding, n→π*at 275 nm) caused a reduction of reflectivity. The peak was broader for p-ETP, as expected from a wider typology of carbonyl species (CO_ox_, –COO^−^).

## 4. Conclusions

Polyaleuritate coatings a few microns thick can be formed on metals commonly used in food canning (Al, TFS and ETP) by the thermal self-esterification of a naturally-occurring fatty polyhydroxyacid (aleuritic acid), directly in air. A kinetic analysis of the reaction revealed the catalytic role of the metal surface, particularly the tin oxide layer on ETP. Although physical characteristics such as adherence to the substrate, glossy appearance and wettability were acceptable, the esterification degree achieved under the preparation conditions used was too low to yield a high molecular polyhydroxyester withstanding the action of solvents. However, the occurrence of an oxidative diol cleavage and further re-esterification side reactions caused a densification of the ester framework and the amorphization of the polyester. A consequence of this exogenous process was a remarkable increment in the resistance to lixiviation. Hence, preparation directly in air, apart from being a cost-attractive process, was very positive for the final properties of the coating, particularly under the promoting effect of the SnO_x_ layer on ETP.

Due to the incomplete information available at the time of this study, the results and conclusions obtained may be considered qualitative and subject to change, according to the specific compositional and morphological features of the metal supports used. 

## Figures and Tables

**Figure 1 polymers-12-00942-f001:**
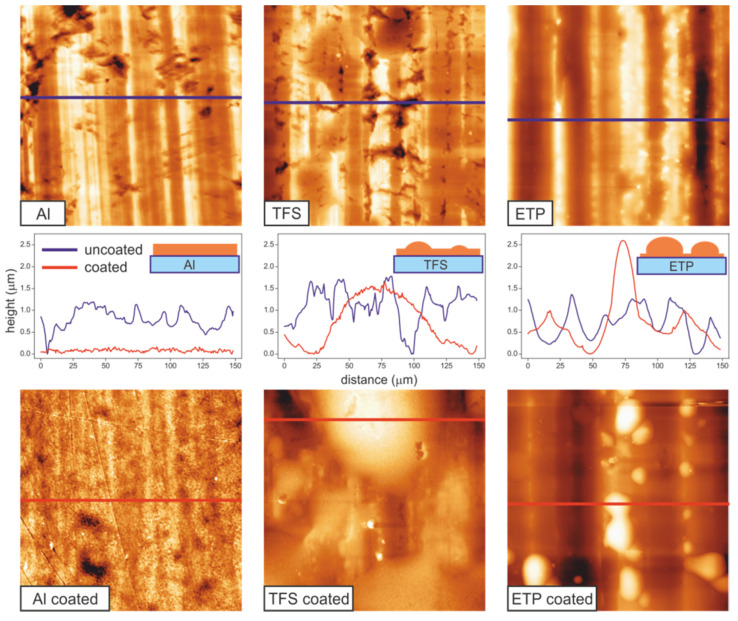
(**top**) 130 × 130 µm^2^ AFM topographic images of bare and (**bottom**) polyaleuritate-coated Al, TFS and ETP supports. Line profiles (**centre**) show the surface roughness modifications upon coating. Different coating patterns are proposed for Al, TFS and ETP, respectively.

**Figure 2 polymers-12-00942-f002:**
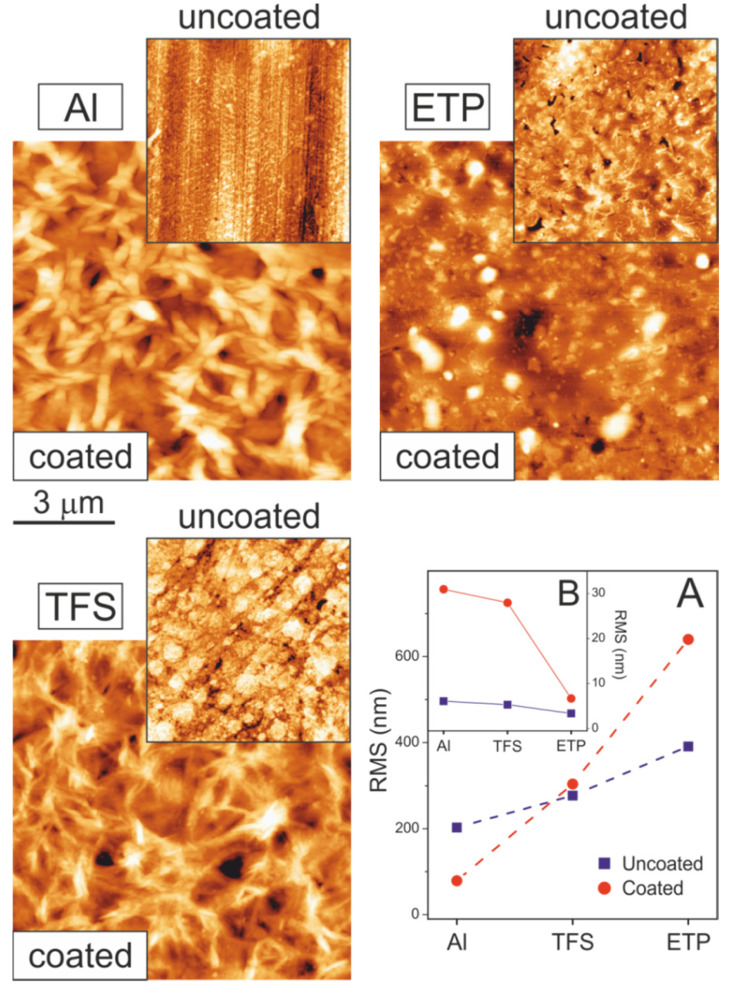
Small range AFM topographic images of bare and polyaleuritate-coated Al, TFS and ETP substrates. The scale bar is applicable to all images. The plot shows the evolution of surface roughness (RMS) obtained from large (**A**) and small (**B**) range AFM images.

**Figure 3 polymers-12-00942-f003:**
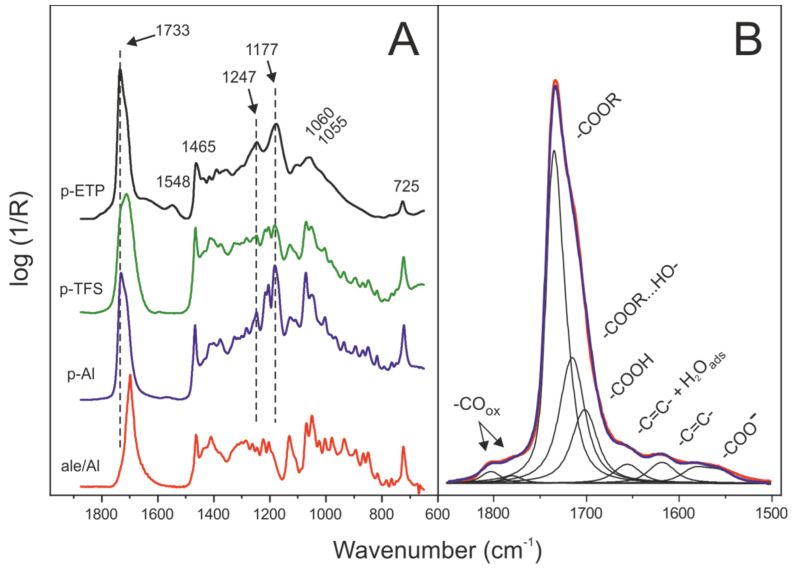
(**A**) Comparative specular reflectance FTIR spectra in the 1875–650 cm^−1^ region of a polyaleuritate film formed on Al (p-Al), TFS (p-TFS) and ETP (p-ETP) substrates at 200 °C for 10 min in air. As a reference, the spectrum of a film of unreacted aleuritic acid on Al (ale/Al) is included. (**B**) Component bands and assignations used for the qualitative and quantitative analyses of specular reflectance FTIR data of polyaleuritate coatings.

**Figure 4 polymers-12-00942-f004:**
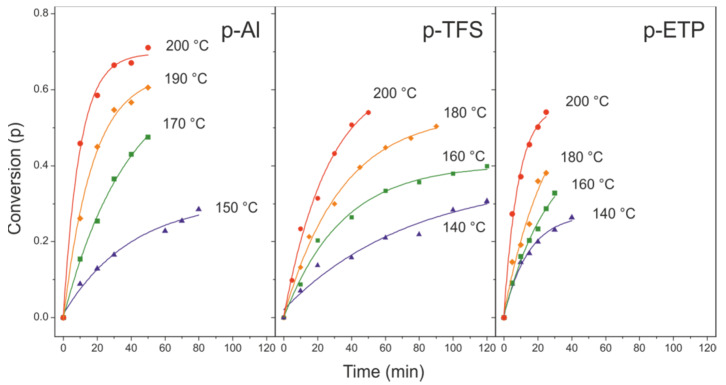
Esterification conversion within polyaleuritate films formed on Al, TFS and ETP supports as a function of temperature and reaction time.

**Figure 5 polymers-12-00942-f005:**
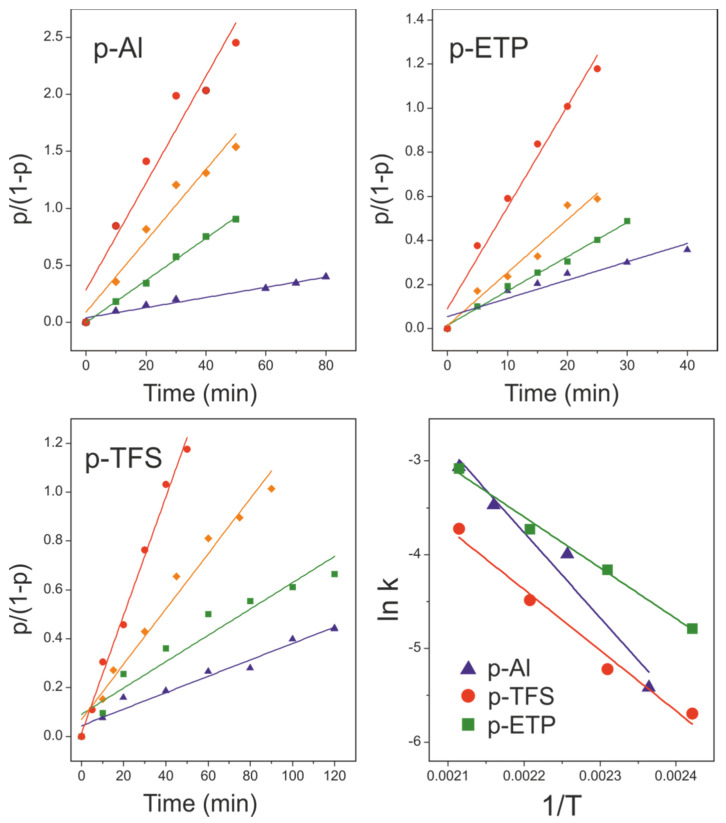
Second order kinetic fittings of esterification degree (p) for polyaleuritate coatings on Al, TFS and ETP substrates (symbol labelling is as in Figure 4). (**Bottom-right**) Arrhenius plots to calculate the activation energy (E_act_) of the process.

**Figure 6 polymers-12-00942-f006:**
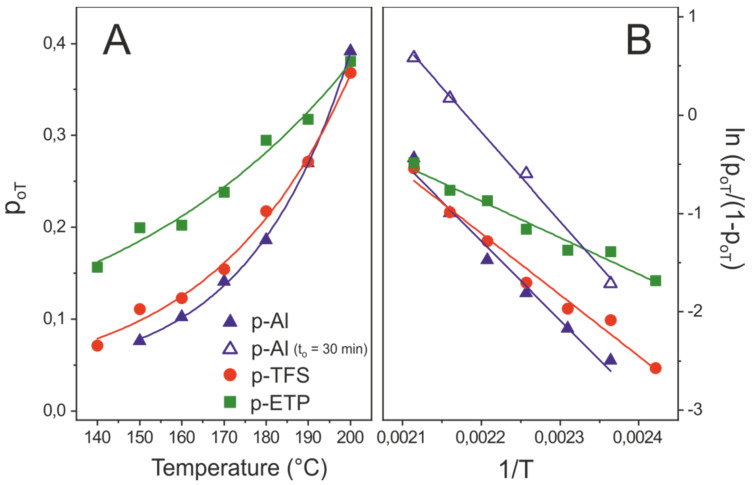
(**A**) esterification degree and (**B**) ln(p_oT_/(1 − p _oT_)) vs 1/T plots for polyaleuritate coatings after 10 min reaction time in air.

**Figure 7 polymers-12-00942-f007:**
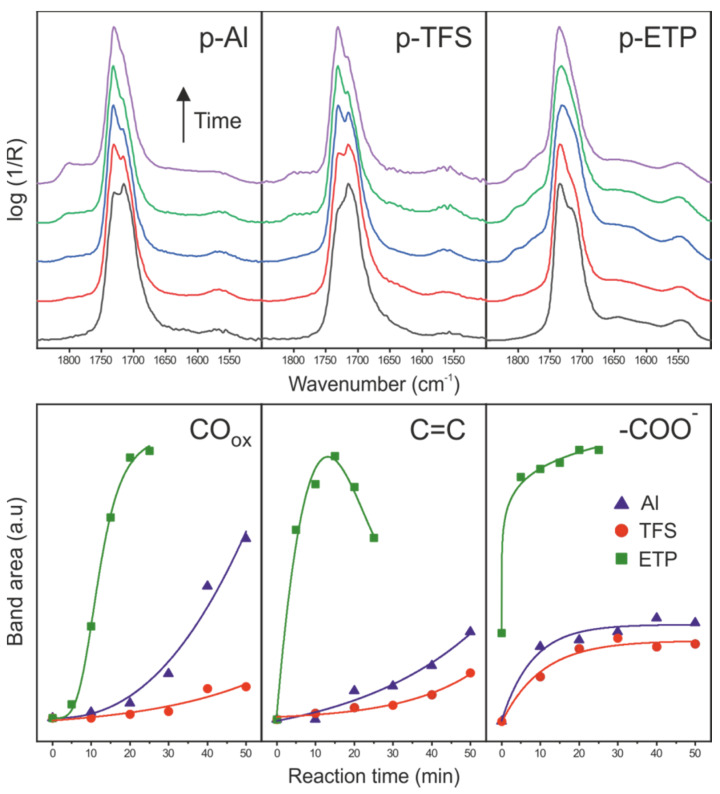
(**top**) ATR-FTIR spectra along the formation of polyaleuritate films on Al, TFS and ETP at 200 °C in air (10–50 min for Al and TFS, 5–25 min for ETP). (**bottom**) Time evolution of CO_ox_, C=C and –COO^−^ species from specular reflectance FTIR spectra.

**Figure 8 polymers-12-00942-f008:**
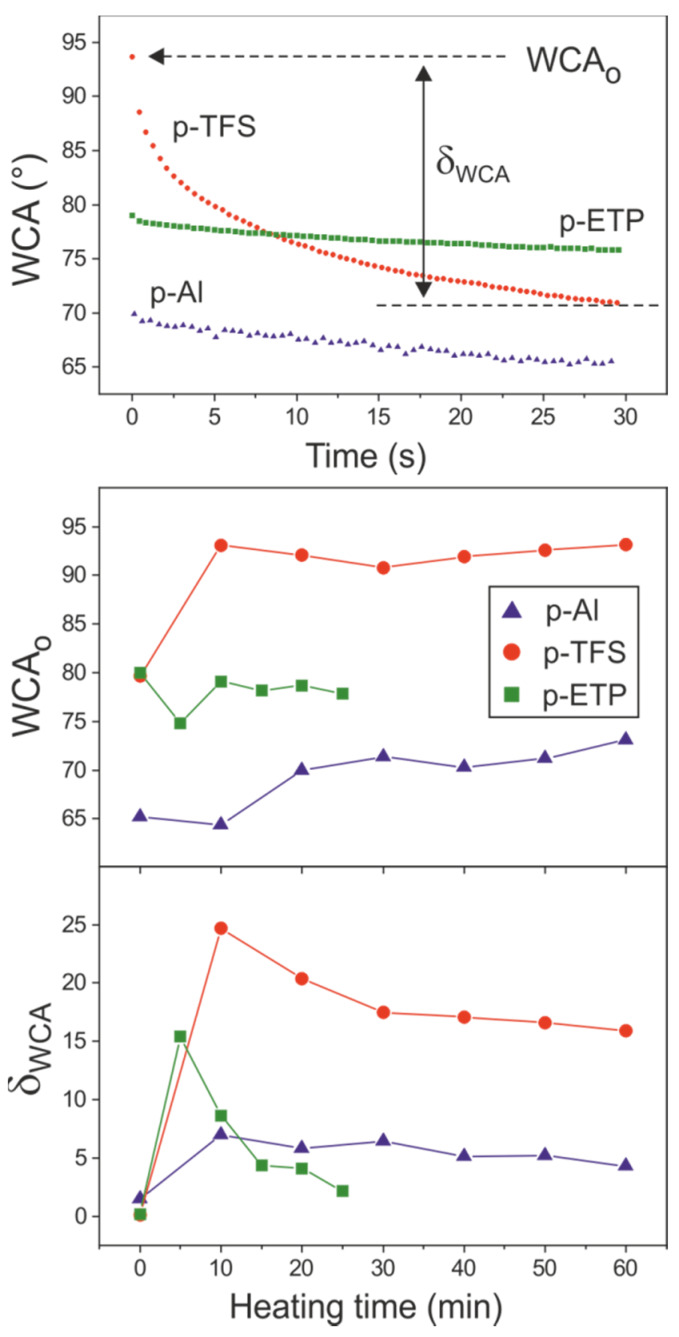
(**top**) Water contact angle (WCA) evolution with contact time on polyaleuritate films obtained at 200 °C for 20 min on Al, TFS and ETP. (**bottom**) Initial WCA value (WCA_o_) and reduction (δ_WCA_) after 30 s (spreading) for coatings prepared at 200 °C and variable time on the three metal supports.

**Figure 9 polymers-12-00942-f009:**
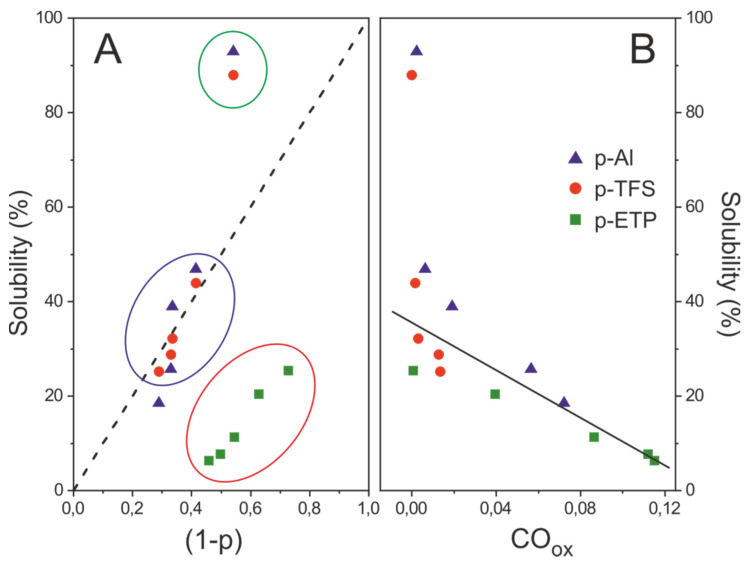
(**A**) Plot of the relative solubility of the polyaleuritate coating vs the fraction of nonesterified aleuritic acid (1 − p). (**B**) Inverse correlation of solubility with the presence of oxidized species formed upon the esterification of aleuritic acid in air. Data correspond to the coatings formed at 200 °C and variable reaction time on the three supports investigated.

**Figure 10 polymers-12-00942-f010:**
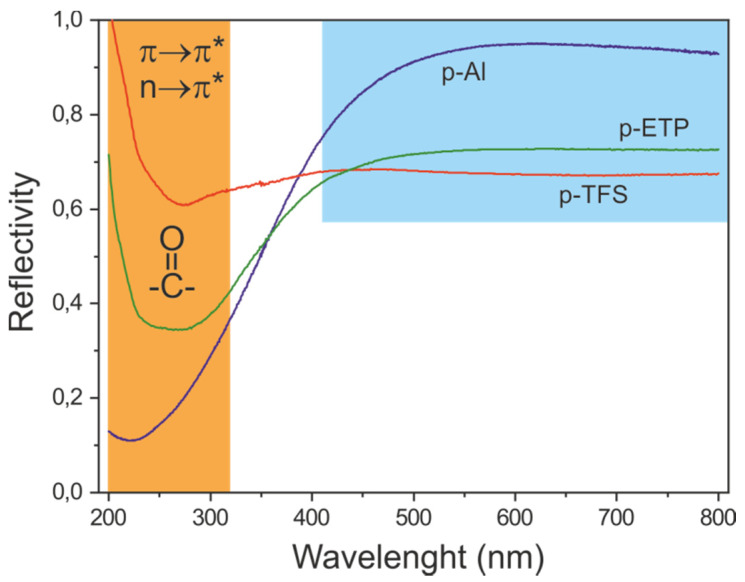
UV-Vis reflectivity of polyaleuritate-coated metal supports. Blue and orange regions correspond to the visible and UV regions, respectively.

**Table 1 polymers-12-00942-t001:** Large and small scale roughness obtained from AFM for bare and polyaleuritate-coated Al, TFS and ETP. The S_F_ parameter is the ratio between the topographic and the X-Y scanned areas; it is related to the specific surface area.

	Large			Small	
	RMS (nm)	h_avg._ (nm)	h_max._ (nm)	RMS (nm)	S_F_
Uncoated					
Al	203 ± 13	1083 ± 86	1676 ± 90	6.1 ± 0.2	1.0782
TFS	277 ± 63	1145 ± 141	2175 ± 128	5.3 ± 0.8	1.0453
ETP	391 ± 116	934 ± 288	1901 ± 387	3.4 ± 0.5	1.0094
Coated					
p-Al	79 ± 80	494 ± 278	1054 ± 319	31 ± 2	1.0368
p-TFS	304 ± 65	1179 ± 335	1936 ± 249	28 ± 2	1.0483
p-ETP	640 ± 102	1415 ± 329	3249 ± 322	6.7 ± 0.5	1.0117

**Table 2 polymers-12-00942-t002:** Pre-exponential factor (A) and E_act_ values for the esterification of polyaleuritate coatings on metal substrates using the time- and temperature-dependent methods.

Support	Time-dependent method		Temperature-dependent method	
	lnA	E_act_ (kJ/mol)	lnA	E_act_ (kJ/mol)
Al	15.5	75 ± 11	13.4 (15.7) *	67 ± 5 (76 ± 3) *
TFS	9.2	54 ± 5	9.5	52 ± 4
ETP	7.7	45 ± 3	4.3	31 ± 2

* for 30 min reaction time.

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
