# Peer review of "Bio-Based Coatings for Food Metal Packaging Inspired in Biopolyester Plant Cutin"

_polymers, 2020, doi:10.3390/polym12040942_

Round 1

Reviewer 1 Report

This is a very interesting and carefully conducted work in order to design a non-toxic coating layer for cans, based on the unique barrier properties of the plant cuticle. The experimental presented appears to be well done and thorough and I agree to be published after some minor corrections.

MATERIALS AND METHODS

  1. Page 2, line 76 and 79. Instead of “XX°C”, should be “XX °C”
  2. Page 2, line 85. Instead of k=0.06, should be k= 0.06
  3. Coating preparation: What is the concentration of aleuritic acid in the sprayed solution? It is a saturated or a measured one?

RESULTS AND DISCUSSION

  1. Page 6, 10,11,12, line 164, 282, 288, 290, 293, 294, 308,326, 332. Instead of “XX°C”, should be “XX °C”
  2. Figure 4, make same correction: Instead of “XX°C”, should be “XX °C”
  3. As the authors mentioned, the metal surface has different interactions in the polymerization reaction of the aleuritic acid. I agree that one of these interactions is in the surface. A hydrophilic or hydrophobic interaction could lead to the large- or small-scale roughness. However, polymerization could be another reason of this. Aleuritic acid could gave a linear polymer if only the carboxyl group and the primary -OH reacts, but what about secondary -OH´s? According with this, for a linear polymer, FT-IR spectra should have only a sharp peak. However, according to fig. 3B, different bands are forming this peak, but there is not band assigned to the secondary esters. Are they no formed in the film?

Author Response

We appreciate the comments, suggestions and corrections made by the referees. Particularly, those of referee 3, which have opened many new perspectives for the improvement of methodology and orientation of future work. We have modified the text accordingly to their evaluation and we hope that the new version fulfill their requirements. In the new version of the manuscript, every change made has been is highlighted in red to facilitate the task of editors and reviewers.

Indications have been addressed as follows:

Referee 1.

We appreciate the kind heading comment made by the reviewer.

Materials and methods

  1. A space after °C has been included all over the text body.
  2. k=0.06 has been corrected to k= 0.06 in line 90.
  3. The concentration of the aleuritic acid used (10 mg/mL) has been included in the 2.1. Coating preparation section, line 80-81:”was dissolved in ethanol (Honeywell, purity >8%) at a concentration of 10 mg/mL and sprayed…

Results and discussion.

1 and 2, a spacing before °C has been added all over the text, including figure captions and in figure 4 labelling.

  1. A new paragraph addressing the influence of hydrophilic and hydrophobic interactions on the surface texture has been added in lines 171-176: ”The textural evolution of the coating is a complex process that results from the dynamic balance between surface-polyaleuritate and polyaleuritate-polyaleurite interactions. In this sense, the hydrophilic forces between the oxide layer and the polar groups of aleuritic acid molecule, as well as, the growing hydrophobic component arising from the formation of the polyaleuritate should be considered. Thus, the formation of large aggregates can be associated with a fast development of a hydrophobic polyaleuritate phase”. As pointed by the referee, the preferential formation of the linear polyester has been indicated and referenced to previous study in lines 193-195: ”The formation of a crystalline polyaleuritate phase is feasible because, in the reaction conditions used, the esterification of the primary hydroxyl to yield a linear polymer is favored [18,19]”. The broadening of the ν(C=O) is justified by the presence of primary esters perturbed by hydrogen-bonding as well as unreacted acid molecules rather than to the low population of secondary esters, lines 200-202: ”The low wavenumber side broadening of the ν(C=O) is due to the perturbation exerted by hydrogen-bonding and to the presence of unreacted acid molecules”.

Reviewer 2 Report

The idea of having biocoatings applied on metals as alternative to epoxi resins is good. The synthesis and characterization of the coating and the application on different  metal surfaces is good. However, the authors emphasize in the introduction the problems from Bisfenol A linked to epoxi resins used as coatingd for cans. They propose to use these new biocoatings on cans. But they don't demonstrate the safety in use, what means the migration tests, the identification of compounds released by the coatings, including quantitative values found. Without these data, they cannot propose the new coatings for being used in contact with food. It is not enough to study the compatibility with solvents. I suggest the authors complete the study with the migration tests, both global and specific migration. Biopolymers also release substances to the food and the requirements applied to them are the same as for conventional polymers.

Author Response

We appreciate the comments, suggestions and corrections made by the referees. Particularly, those of referee 3, which have opened many new perspectives for the improvement of methodology and orientation of future work. We have modified the text accordingly to their evaluation and we hope that the new version fulfill their requirements. In the new version of the manuscript, every change made has been is highlighted in red to facilitate the task of editors and reviewers.

Indications have been addressed as follows:

Referee 2.

We absolutely agree with the reviewer assessment about the safety compliance of these coatings before being proposed as a viable solution for food canning. We also consider that not only rigorous migration tests, but also a more exhaustive analysis of adherence, chemical inertness and mechanical performance should be carried out. However, in this paper we wanted to report a fundamental study covering preliminary aspects such as the feasibility of preparation by direct heating in air and in the absence of undesired solvents and catalysts. We also intended to explore the peculiarities of the preparation and the coating properties on the most common metals and alloys used in the canning industry. We also considered of scientific interest to gather kinetic information about the self-esterification reaction as well as the catalytic effect of the oxides at the metal surface. With such knowledge, preparation protocols can be better designed, for instance, the need for a surface activation on less effective substrates such as Al and TFS via SnOx deposition or the addition of compatible cross-linkers to speed up the amorphization process.

We think that food safety evaluation is among the final stages of coating design and should be performed on much more optimized systems, which is not the case here. But we consider that the study carried out is of interest for the scientific community on the topic.

Reviewer 3 Report

Introduction

Line 35 .... Bisphenol is practically excluded from varnishes and lacquers according to ISO 14000. The references in this regard are from 2008-2013, a problem that is decreasing due to regulations.

line 57-59 .... objective is not clear regarding the natural innovative coating ... is it chemically synthesized ?, in mixture with the mentioned polymer?, .... improve wording for the reader.

line 66-67 ... it is necessary to specify the chemical composition of the 3 types of packaging materials (% Al and additive, low tin ETP, layers, etc., TFS and ETP the steel composition, for example). the 3 materials?

Materials and methods

line 73-74 .... how did you ensure in cleaning not to remove the chrome substrates that are of a thickness of the nanometric order?

line 76 ... aim to preheat Al? Was it characterized before and after preheating to assess its surface condition?

line 81-82 ... the measurement of adherence is rudimentary and qualitative. You could try a T-peel test that is quantitative and allows you to get the work and energy of adhesion, this is key to durability and pore control.

Textural characterization

Looking for a specific texture? This because the mechanical conformation of the container of the three materials Al, ETP and TFS depends on a type of texture.

Chemical analysis of coatings.

line 91 ... more details are required of the technical operating conditions of the FT-IR, under what conditions?

 Wettability and solubility measurements.

Were standards used for measurement in the materials? How was the value of the contact angle of the samples measured?

UV-vis reflection spectra

What is the justification for including this test, since the new coating is internal to the container ... or also external?

Results and Discussion.

line 113 ... what are the oxidation states of the aluminum layers or substrates?

line 112-115 TFS should have two substrates a very thin Cr zero and a nanometric oxide, were they not characterized? On the other hand, ETP is steel with FeSn2 and a tin oxide whose function is anodic to protect the steel in the degradation or attack of an electrolyte

... was not characterized? This is key to the barrier function of the new coating. The pores facilitate lectrochemical interactions between the ETP substrates. Analyze this suggestion.

line 118. texture should not be confused with surface roughness are different concepts. Check wording.

120-123 This required a more detailed analysis. The structures are rather homogeneous and not granular in ETP and TFS, they are continuous substrates. It is important to verify that no substrates were characterized under the protective coating, which could falsify the observations and results.

line 134 the result of the topography changes also depend on the surface quality of the base metal Al, ETP and TFS. Therefore the discussion should be based on patterns and with coating. Discuss this aspect.

line 143-157 The results different surface states for the materials. It is important to know that a finishing oil (such as DOS for example) is applied to food packaging after the protective coating. Therefore, it could be explained to what extent these topographic differences can influence the "functionality of the material" according to the characterization .

line 171-172 ... the wavelength at which there are crystallinity changes would be convenient. Keep in mind that an amorphous phase favors adherence and a crystalline phase favors resistance to abrasion (food movement) and mechanical resistance of the finishing substrate.

line 187-190 Normally the carbonyl group binds with the chromium oxide of the TFS. The determination or analysis of hydrogen contributes to the breakage of the weaker van der waals layers that will facilitate the adhesion of substrates.

line 208. The curves are quite representative, a suggestion in Fig 28 would be to obtain the coefficient of determination "to have the influence of the variables involved.

line 228. which means that the results are in "good agreement", justify.

line 300. Conclusive results. What is the difference in permeability in this case?

line 308. Ethanol proves to be a good solvent for aleurites, does this mean that all three materials can be recyclable after use, and would they contribute to the green economy? To what degree does the operational process of dissolving with methanol imply specific temperature control conditions? Does this apply to the quality control stage of the materials tested with the new coating?

Author Response

We appreciate the comments, suggestions and corrections made by the referees. Particularly, those of referee 3, which have opened many new perspectives for the improvement of methodology and orientation of future work. We have modified the text accordingly to their evaluation and we hope that the new version fulfill their requirements. In the new version of the manuscript, every change made has been is highlighted in red to facilitate the task of editors and reviewers.

Indications have been addressed as follows:

Referee 3.

Introduction

  1. Bisphenol is practically excluded from varnishes and lacquers according to ISO 14000. The references in this regard are from 2008-2013, a problem that is decreasing due to regulations…... The referee is right in this assertion, current regulations are banning the use of BPA in can linings. Indeed, can manufacturers associations claimed in 2018 that BPA has been removed from more than 90% of their production. However, consumer associations still report the presence of this compound in canned food. In 2016 (only 3, 4 years ago), brands like Campbell´s, Del Monte, Nestlé, McCormick & Company, General Mills still were reported to use BPA lining in 50% - 90% of their cans. Some of them indicated their intention to create BPA-free containers by the end of 2017, but others had, at this time, no timeline to provide an acceptable alternative.http://www.toxicfoodcans.org/wp-content/uploads/2016/03/BPA-BuyerBeware.pdf. Taking into account the discrepancies between declarations of can producers and consumer reports, and the short time frame in between, we consider that the concern about the presence of BPA in canned food is still a latent issue. We have made reference to BPA as the case that triggered the worry about toxic substances migrating from the can lining and the need for a safer alternative formulation. The Introduction section has been modified to somehow reduce the focus on BPA and extend it to other concerns that may arise in the years to come. Lines 34 to 55 of the new version of the manuscript have been modified.
  2. .... objective is not clear regarding the natural innovative coating ... is it chemically synthesized ?, in mixture with the mentioned polymer?, .... improve wording for the reader. Indeed, the paragraph is confusing, we have rewritten line 56 to 64 as follows: Our approach to design a non-toxic coating layer for cans is based on the observation of the unique barrier properties of the plant cuticle. The cuticle is the outermost membrane that protects the epidermis of fruits, leaves and non-lignified stems of higher plants [17]. The main component of the cuticle is cutin (up to 80% w/w), a non-toxic, biodegradable, hydrophobic, insoluble and infusible amorphous biopolyester mostly made of interesterified C16 polyhydroxy acids. The goal is to confirm whether a synthetic C16 polyhydroxyester resembling cutin may be developed as an effective coating for food cans. For this purpose, a C16 polyhydroxyacid such as aleuritic acid (9,10,16-trihydroxyhexadecanoic acid) has been polymerized”.
  3. ... it is necessary to specify the chemical composition of the 3 types of packaging materials (% Al and additive, low tin ETP, layers, etc., TFS and ETP the steel composition, for example). the 3 materials?. As indicated in the text, metals were a donation from AkzoNobel and we did work with them under the consideration that they were common materials used for can fabrication and the same supports used by their research facilities to test their lacquers. Unfortunately, we have no further information from the manufacturers. We did conduct EDX analysis and, in the case of Al only traces of Si (0.35% At.) and Fe (0.26% at.) were found. FTIR showed no organic compound on the surface. Only Fe and Cr were detected in TFS, chromium content was 0.8% at. and in ETP only Fe, O and Sn were observed (Sn 32% at.). Those results are now included in the 1 Texture of supports and coatings section. Unfortunately, at this point we have no further information regarding the layer structure or the type of iron/steel matrix.

Materials and methods

  1. .... how did you ensure in cleaning not to remove the chrome substrates that are of a thickness of the nanometric order? The cleaning process was indicated by AkzoNobel staff and consisted in a few seconds low power sonication at RT in a solution of glassware soap, followed by rinsing with water and ethanol. We don´t think this is aggressive enough to remove the chromium oxide from the surface of the TFS substrate. The above mentioned EDX analysis was carried out on samples cleaned by the indicated protocol and Cr was detected on TFS.
  2. ... aim to preheat Al? Was it characterized before and after preheating to assess its surface condition?. Heating was necessary because the ethanol used as a solvent did wash the aleuritic deposit under the stream coming from the airbrush. We found that the fast ethanol evaporation on hot supports solved this problem and quite smooth coatings were obtained, very likely also helped by the fusion of the aleuritic acid. We consider that ~100 °C is not a temperature high enough to cause a meaningful alteration of the supports, particularly when they were heated at much higher temperature along the polymerization of the hydroxyacid.
  3. ... the measurement of adherence is rudimentary and qualitative. You could try a T-peel test that is quantitative and allows you to get the work and energy of adhesion, this is key to durability and pore control… Indeed thumb nail scratching and adhesive tape testing, though widely used for routine quality control, is a quite rudimentary methodology. However, it must be taken into account that this is a preliminary study mainly focused on fundamental aspects such as the viability of the direct self-esterification of the fatty hydroxyacid in air, the kinetics of the reaction and the role of the external oxide layer on the final properties of the coating. With this information, preparation protocols would be designed and, if successful, a more exhaustive study addressing migration, adherence, chemical resistance and mechanical performances would be necessary. But, at this stage, we only needed qualitative information to tell us if we were going in the right direction.

Textural characterization

  1. Looking for a specific texture? This because the mechanical conformation of the container of the three materials Al, ETP and TFS depends on a type of texture... Not really, we just wanted to check if the substrate texture exerted any influence on the spreading of the polyaleuritate layer and in the kinetics of the reaction. Also to characterize the roughness of the coating, which is relevant to migration and to microorganism adhesion and proliferation studies.

Chemical analysis of coatings

  1. ... more details are required of the technical operating conditions of the FT-IR, under what conditions?...The section now includes some operating details and is rewritten as follows “The chemical analysis of coatings was performed by FT-IR spectroscopy using a specular reflectance accessory (Smart SpeculATR, Thermo Scientific) coupled to a Nicolet iS50 spectrometer equipped with a DLaTGS detector. The accessory was continuously purged with dry N2 to reduce the contribution of ambient CO2 and water. Specular reflectance is a very suitable mode for large areas and deep sampling of polymer coatings thicker than 1 µm on metals. In this case, the analysis area was ~1 cm2 which ensured high signal levels and sample representativeness. Fifty scans were accumulated at 4 cm-1 resolution and clean metal supports were used as background. Signal intensity is quantitative when expressed as the logarithm of the inverse of the relative reflectance (log (1/R)). Data acquisition, processing and band fitting was performed with the OMNIC 9 (Thermo Scientific) software package”.

Wettability and solubility measurements

  1. Were standards used for measurement in the materials? How was the value of the contact angle of the samples measured?.... We used no specific standard procedures (i.e. ASTM or ISO) for solubility and WCA measurements. In this later case we followed the set of recommendations provided by the manufacturer: the use of Milli-Q grade water, a drop volume between 2 and 5 µL, the optical calibration of the camera with a sphere of known dimensions and measuring times below 30 seconds to avoid the influence of the liquid evaporation. In the paper we have not tabulated WCA values, we just plotted them in figure 8. The experimental of WCA has been completed with a couple of sentences: “..the contour was recorded for 30 s at 12 fps. Contact angle was measured at both sides of the drop contour and averaged. Frames with left and right values differing more than 2° were rejected”.

UV-vis reflection spectra

  1. What is the justification for including this test, since the new coating is internal to the container ... or also external?... As we mentioned in the text: The interaction of the inner protective coating with visible light is irrelevant from the point of view of the preservation of food in metal containers” though we are aware that certain colorations and lack of uniformity are aspects that cause a negative impression to consumers. Our motivation to carry out such measurements is simply from an esthetic point of view. We consider that the potential of these coatings is as internal varnishes. There are already many other suitable options for external ones.

Results and discussion

  1. ... what are the oxidation states of the aluminum layers or substrates?. TFS should have two substrates a very thin Cr zero and a nanometric oxide, were they not characterized? On the other hand, ETP is steel with FeSn2 and a tin oxide whose function is anodic to protect the steel in the degradation or attack of an electrolyte. ... was not characterized? This is key to the barrier function of the new coating. The pores facilitate lectrochemical interactions between the ETP substrates. Analyze this suggestion. Unfortunately, we did not carry out XPS measurements to stablish oxidation states of surface layers, neither any additional measurement to set the composition depth profile of the supports. As mentioned above, we made a survey EDX analysis to confirm the presence main components. We relied on the representativeness of samples provided by AkzoNobel for those used in actual production lines. Be certain that this is a precaution we will take from now on.
  2. texture should not be confused with surface roughness are different concepts. Check wording. That is right. For this reason, the heading of the 3.1. section has been modified as well as the caption of table 1.
  3. 120-123 This required a more detailed analysis. The structures are rather homogeneous and not granular in ETP and TFS, they are continuous substrates. It is important to verify that no substrates were characterized under the protective coating, which could falsify the observations and results… Indeed the three supports are quite flat. Actually, in the corresponding small range AFM images in figure 2, height values are an order of magnitude smaller that dimensions in the X,Y plane. The term “granular” is commonly used in AFM when the edges of motifs are visible and define recognizable features, as in TFS. It does not necessarily means protruding grains. The same is applicable to ETP (which is even flatter). In this latter cases surface structures were defined as “interconnected”. The texture of coated Al and TFS is quite different from those of the bare supports, which consistently indicates that is the coating what is being imaged. The textural variation in ETP is more subtle and doubts may arise. This is why full coverage of ETP is presumed from large scale images, as indicted in lines 155 to 159 in the new version of the manuscript: “Though the full coverage of Al and TFS supports can be inferred from the long range textural analysis, it is not possible to asset whether this is the case for ETP. Line profiles (Figure 1) show that the underlying texture of ETP is mostly unaltered by the coating. However, the size of the globules does not seem to be enough to account for the mass of a 2-3 µm thick flat coating and, consequently, the development of a very thin layer at the background regions of ETP is presumed”.
  4. line 134 the result of the topography changes also depend on the surface quality of the base metal Al, ETP and TFS. Therefore the discussion should be based on patterns and with coating. Discuss this aspect…. We don´t quite understand this comment. If the question is whether the texture modifications observed are caused by non uniformity of the supports rather than to the coating itself, I has to be taken into account that two to three different samples, at four locations each, have been analyzed for each support and preparation using both the large and small range scanner and topographic patterns obtained are highly repetitive. To clarify this point a sentence is added at the end of the AFM experimental section: “For representativeness, two to three preparations per sample were imaged at 4 distant points using both the large and small scanners”. If the question is that the topography of the coated samples is influenced by the one of the underlying supports, it is truth for large range analysis but, such influence is palliated by the obtaining of small range images.
  5. line 143-157 The results different surface states for the materials. It is important to know that a finishing oil (such as DOS for example) is applied to food packaging after the protective coating. Therefore, it could be explained to what extent these topographic differences can influence the "functionality of the material" according to the characterization… In that case many of the properties of the coating would change, for instance the water contact angle and solubility. However, in this paper, topography is related to surface wetting as well as chemical and structural aspects of the esterification rather than becoming a parameter to tune up.
  6. line 171-172 ... the wavelength at which there are crystallinity changes would be convenient. Keep in mind that an amorphous phase favors adherence and a crystalline phase favors resistance to abrasion (food movement) and mechanical resistance of the finishing substrate… As mentioned in the previous point, the possibility of controlling crystallinity was an aspect we did not contemplate and may be reconsidered in future work. In this sense, the atmosphere control (reducing the concentration of oxygen) would play an important role. We appreciate the suggestion.
  7. line 187-190 Normally the carbonyl group binds with the chromium oxide of the TFS. The determination or analysis of hydrogen contributes to the breakage of the weaker van der waals layers that will facilitate the adhesion of substrates…. We have not observed any carbonyl band in the FTIR spectra that wasn´t characterized previously on free-standing films. Particularly, the 1715 cm-1 contribution has been extensively observed and assigned to the perturbation by hydrogen bonding (excess of hydroxyls with respect to –COOH groups).We expected that a carbonyl interacting with a metal or and oxide may show peaks way above the main ester band at 1733 cm-1. Those at 1800 cm-1 and 1775 cm-1 are fingerprints of acyl peroxides and peroxyesters.
  8. line 208. The curves are quite representative, a suggestion in Fig 28 would be to obtain the coefficient of determination "to have the influence of the variables involved…We don´t quite understand which set of data/curves the referee is making reference to. Is it the fitting of the ν(C=O) region?. We suspect there is some mismatch in line numbering between the original and edited versions of the manuscript that is making difficult to answer some of the questions of the referees.
  9. line 228. which means that the results are in "good agreement", justify… We presume the referee is making reference to the values of A and Eact obtained by the time-dependent and the temperature-dependent methods. For clarity, we have modified the paragraph as follows: “Pre-exponential factors and activation energy values obtained by this second method are also included in Table 2. Reliability of tabulated A and Eact values is supported by the good agreement between results obtained by both the time-dependent and temperature-dependent methods. The temperature-dependent method is faster because requires the preparation of a lower number of samples but demands the previous knowledge of the empirical reaction order
  10. line 300. Conclusive results. What is the difference in permeability in this case?...We made a mistake when drop spreading was associated to drop penetration within the coating. As is written in the text, the evolution of dWCA can be erroneously and univocally related to water permeability. The right concept is that spreading depends on the physical and chemical interaction between the liquid and the coating. To correct this error. The paragraph has been changed as follows: “Both, the low esterification degree (Figure 4) and the porous texture (Figure 2) may contribute to an intense interaction between water and the coating on TFS”.

line 308. Ethanol proves to be a good solvent for aleurites, does this mean that all three materials can be recyclable after use, and would they contribute to the green economy? To what degree does the operational process of dissolving with methanol imply specific temperature control conditions? Does this apply to the quality control stage of the materials tested with the new coating?....Solubility tests made with ethanol simply intended to qualitatively reflect the resistance of the coating to solvents though, certainly, it may become a quality control parameter. On the other side, and, as concluded in this study, the best performances are observed for the amorphous layer, which is quite insoluble. The coating would be burned out along the recuperation of metals by melting because the thermal decomposition of polyaleuritate in air is completed around 375 °C.

Round 2

Reviewer 2 Report

The manuscript has been considerably improved and I think that it can be published. The authors answer the questions and solve the problems.

Author Response

We appreciate the fast response of the reviewer and the time spent on the revision of our paper.

Reviewer 3 Report

The responses to my observations are adequate and clarify the manuscript, considering the focus and objectives of the investigation. I confirm that it is an interesting work and that it generates new knowledge.

Details pending of their answers, for me they are in the limit of the realized thing and of the improvement of the future work of the research line.

However, in my opinion, it is necessary to ask or find out from the supplier of the materials some specifications for publication. This, because if the material is unknown in its chemical composition in the substrates, the purpose of the investigation loses relevance.

It is key, since the new coating had a specific performance that cannot be generalized for materials in the packaging industry.

They should obtain from the company that donated the materials the following (which was not answered by the authors):

TINPLATE. If the steel is Type L or MR
TINPLATE TIN COATING. Type E or D designation
TFS. If the chrome plating process is "one or two steps".
ALUMINUM. The answer clarifies its type.

If it is not possible to obtain this information, in the introduction and discussion they should specify the limitations of the study of the coating with respect to the surface due to the lack of composition mentioned substrates of the materials.

Author Response

We appreciate the fast response and dedication of the referee to review our paper. Corrections, suggestions and indications are really relevant and worth of consideration in the new text. As indicated, we have mentioned the lack of specific composition and manufacturing details of ETP and TFS metal substrates in the experimental section (lines 76-78) as follows" ...but specific information about their composition and manufacturing process is missing. For instance, we ignore the type of steel matrix (L or MR) and tin coating (E or D) in ETP substrates and the chromium plating (one or two steps) in TFS.". Consequently, a final statement indicating the precautions that should be taken when interpreting our results and conclusions is added at the end of the Conclusion sections: "...Due to the uncomplete information available at the time of this study, results and conclusions obtained may be considered as qualitative and subjected to modulation by specific compositional and morphological features of the metal supports used".

We hope such modification would account for the requirements of the referee.